# Non-Invasive Evaluation of Acute Effects of Tubulin Binding Agents: A Review of Imaging Vascular Disruption in Tumors [note 1]

**DOI:** 10.3390/molecules26092551

**Published:** 2021-04-27

**Authors:** Li Liu, Devin O’Kelly, Regan Schuetze, Graham Carlson, Heling Zhou, Mary Lynn Trawick, Kevin G. Pinney, Ralph P. Mason

**Affiliations:** 1Department of Radiology, University of Texas Southwestern Medical Center, Dallas, TX 75390, USA; Li.Liu@UTSouthwestern.edu (L.L.); Devin.OKelly@UTSouthwestern.edu (D.O.); Regan.Schuetze@UTSouthwestern.edu (R.S.); helingzhou7@gmail.com (H.Z.); 2Department of Chemistry and Biochemistry, Baylor University, Waco, TX 76798, USA; Graham_Carlson@baylor.edu (G.C.); Mary_Lynn_Trawick@baylor.edu (M.L.T.); Kevin_Pinney@baylor.edu (K.G.P.)

**Keywords:** imaging, bioluminescence, photoacoustics, magnetic resonance imaging, vascular disrupting agents, inhibitors of tubulin polymerization, breast cancer, kidney cancer, lung cancer, combretastatins

## Abstract

Tumor vasculature proliferates rapidly, generally lacks pericyte coverage, and is uniquely fragile making it an attractive therapeutic target. A subset of small-molecule tubulin binding agents cause disaggregation of the endothelial cytoskeleton leading to enhanced vascular permeability generating increased interstitial pressure. The resulting vascular collapse and ischemia cause downstream hypoxia, ultimately leading to cell death and necrosis. Thus, local damage generates massive amplification and tumor destruction. The tumor vasculature is readily accessed and potentially a common target irrespective of disease site in the body. Development of a therapeutic approach and particularly next generation agents benefits from effective non-invasive assays. Imaging technologies offer varying degrees of sophistication and ease of implementation. This review considers technological strengths and weaknesses with examples from our own laboratory. Methods reveal vascular extent and patency, as well as insights into tissue viability, proliferation and necrosis. Spatiotemporal resolution ranges from cellular microscopy to single slice tomography and full three-dimensional views of whole tumors and measurements can be sufficiently rapid to reveal acute changes or long-term outcomes. Since imaging is non-invasive, each tumor may serve as its own control making investigations particularly efficient and rigorous. The concept of tumor vascular disruption was proposed over 30 years ago and it remains an active area of research.

## 1. Introduction

Solid tumor growth beyond about 1-3 mm in diameter depends extensively on angiogenesis initiating neovasculature for the supply of nutrients and oxygen [1]. However, tumor neovasculature is abnormal, in terms of both structure and function, and has been proposed as a specific target for therapeutic intervention [2,3,4,5,6]. Notably, tumor endothelial cells undergo rapid proliferation and vessels generally lack pericyte coverage [7,8]. Two types of therapy have been proposed to target tumor-associated vasculature: angiogenesis inhibiting agents (AIAs) inhibit the development of blood vessels a priori [9], while vascular disrupting agents (VDAs) specifically target existing neovasculature [4,5,10,11]. Many small-molecule VDAs interact with the tubulin-microtubule protein system including the well-characterized vinca alkaloid and colchicine binding sites, which are located separately on the αβ-tubulin heterodimer [12,13]. In the late 1970′s, Pettit and co-workers discovered the combretastatins in the South African bush willow tree, *Combretum caffrum*, of which combretastatin A-1 (CA1) [14] and combretastatin A-4 (CA4) [15] are two of the most potent compounds, each exhibiting pronounced biological activity as inhibitors of tubulin polymerization and as selective VDAs. VDA activity results from microtubule disruption in activated endothelial cells, which initiates a signaling pathway characterized by profound cytoskeletal and morphological changes [16,17]. Consequently, endothelial cells round up, leading to enhanced vascular leakage, and detachment from each other and from the underlying substratum to clog the tumor blood vessels [18]. Direct vascular disruption is predicted to cause massive downstream starvation and hypoxiation, thereby potentiating the local effect and generating extensive necrosis [4] (Figure 1).

While the concept of vascular targeting was proposed some 30 years ago, we note a dramatic recent increase in interest, particularly efforts to develop next generation agents: 6700 Web of Science citations of “combretastatin” in 2020 represented a 15-fold increase since 2000. Several VDAs have been evaluated in clinical trials, though to date none has received FDA approval [19,20,21,22]. Table 1 outlines the status of several VDAs and lists clinical trials that included substantial imaging. Several VDAs (CA4P, DMXAA and ZD6126) underwent Phase 2 clinical trials and CA4P and AVE8062 reached Phase 3 [21,23]. CA4P was granted the status of an orphan drug by the European Medicines Agency (EMA) and Federal Drug Administration (FDA) [19]. Several clinical trials are listed at ClinicalTrials.gov, but only one is currently active: “Modulation Of The Tumour Microenvironment Using Either Vascular Disrupting Agents or STAT3 Inhibition in Order to Synergise With PD1 Inhibition in Microsatellite Stable, Refractory Colorectal Cancer (MODULATE)“ which examines BNC105P and is organized by the Australasian Gastro-Intestinal Trials Group. To date imaging, as a biomarker, has been incorporated into relatively few clinical trials and the potential value and shortcomings were discussed extensively by O’Connor et al., particularly regarding the parameters measurable by DCE-MRI [24].

It is recognized that VDAs are ineffective as monotherapies, since a thin peripheral rim of cells, thought to receive nutrients from the host vasculature survives, even after destruction of the tumor vasculature. While the tumor center may necrose, the rim often repopulates rapidly. As such, several VDAs have been tested in combination with additional therapies [11,41], including radiotherapy [41,42,43,44,45,46], antiangiogenic agents (such as bevacizumab) [47,48], traditional cytotoxic chemotherapy (e.g., carboplatin, paclitaxel) [41,49,50,51,52,53,54] and recently immunotherapy [55,56]. There is a current resurgence of interest in VDAs and frequent reports describe novel agents, many based on the colchicine/combretastatin motif [57,58,59,60,61,62,63,64,65,66,67,68,69,70,71,72] (Figure 2A). These molecules are typically hydrophobic and are modified as phosphate prodrugs to enhance aqueous solubility and allow ease of delivery. The phosphates are intrinsically less active, particularly in terms of tubulin binding, as assessed in cell free assays [73,74], but non-specific phosphatases are abundant in cells providing rapid release of the active agents [75]. Recent reports have explored the encapsulation of combretastatin and DMXAA in targeted nanoparticles, sometimes in combination with co-encapsulated chemotherapy or anti-angiogenesis drugs to enhance tumor retention and prolong effective release [76]. While combretastatins have seen substantial progress in clinical development (Phase I–III clinical trials, Table 1), several other molecular structures can selectively lead to destruction of tumor vasculature and examples are shown in Figure 2B. Notably, arsenic trioxide (Trisinox; ATO) is used clinically to treat promyelocytic leukemia and has been shown to cause vascular disruption in solid tumors [77,78,79], though at low doses it interferes with mitochondrial activity and actually increased tumor oxygenation, as revealed by ESR and ^19^F MR oximetry [80]. Selective vascular destruction has also been achieved using antibody targeted tissue factor (anti-VCAM-1.TF) [6,81] and physical approaches based on photodynamic therapy [82,83,84,85], microwave heating [86] or high dose radiation [87] potentially enhanced with high-z nanoparticles [88]. The application of imaging for non-invasive assessment of vascular disrupting agent activity is presented in Table 2.

**Table 2 molecules-26-02551-t002:** Pre-clinical Imaging of VDAs.

Agent	Imaging Modality	Tumor Type	References
CA4P	^a^ BLI, ^b^ MRI, ^c^ MSOT/PAT, ^d^ PET/CT, ^e^ EPR, ^f^ US, ^g^ SPECT	Breast, liver, colorectal, bladder, pancreatic, prostate, lung, melanoma	^a^ [5,89,90,91,92,93] cf. Figures 3 and 4 ^b^ [5,27,91,93,94,95,96,97,98,99,100,101,102,103,104,105,106,107] ^c^ [92] cf. Figures 8 and 9 ^d^ [105,108], ^e^ [95] ^f^ [109,110,111,112], ^g^ [113]
CA1P	^a^ BLI, ^b^ MRI, ^c^ MSOT, ^d^ PET/CT, ^f^ US	Colorectal, H&N, breast	^a^ [5,114,115] ^b^ [103,105,114,116,117] ^c^ [118,119] ^d^ [105], ^f^ [5,91]
BNC105P	BLI	Kidney	[120]
AVE8062	BLI, MRI, FDG-PET, CE-US	Colon, Ovarian, H&N	[121,122,123]
NPI2358	DCE-MRI	Breast, sarcoma	[124]
ZD6126	DCE-MRI, BOLD MRI	Colon, breast, prostate, fibrosarcoma	[11,37,125,126,127,128]
BPR0L075	BLI	Breast	[129]
EPC2407	BOLD MRI, DCE-MRI, MSOT, BLI, US	Head & Neck, glioma, prostate	[130,131,132]
DMXAA	^a^ BLI, ^b^ MRI, ^c^ MSOT, ^d^ FDG-PET	Breast, colorectal, glioma, kidney, H&N	^a^ [133,134], ^b^ [91,96,134,135,136,137,138], ^c^ [133,139,140] ^d^ [141]
OXi8007	BLI, MRI, US	Breast, prostate	[17,74,93] cf. Figures 5, 9 and 11
C118P	MRI	Liver (rabbit)	[142]
ABT-751	MRI	Glioma (rat)	[143]
CA4P analogs	BLI, US	Breast, prostate, lung	[66,74,144,145,146,147]
Targeted prodrugs	BLI	4T1 breast	[148]
Nanoparticles	MRI, optical, MSOT	4T1, MCF-7 breast	[149,150,151,152,153]
Conjugates	MSOT	Colon	[154]

**Figure 2 molecules-26-02551-f002:**
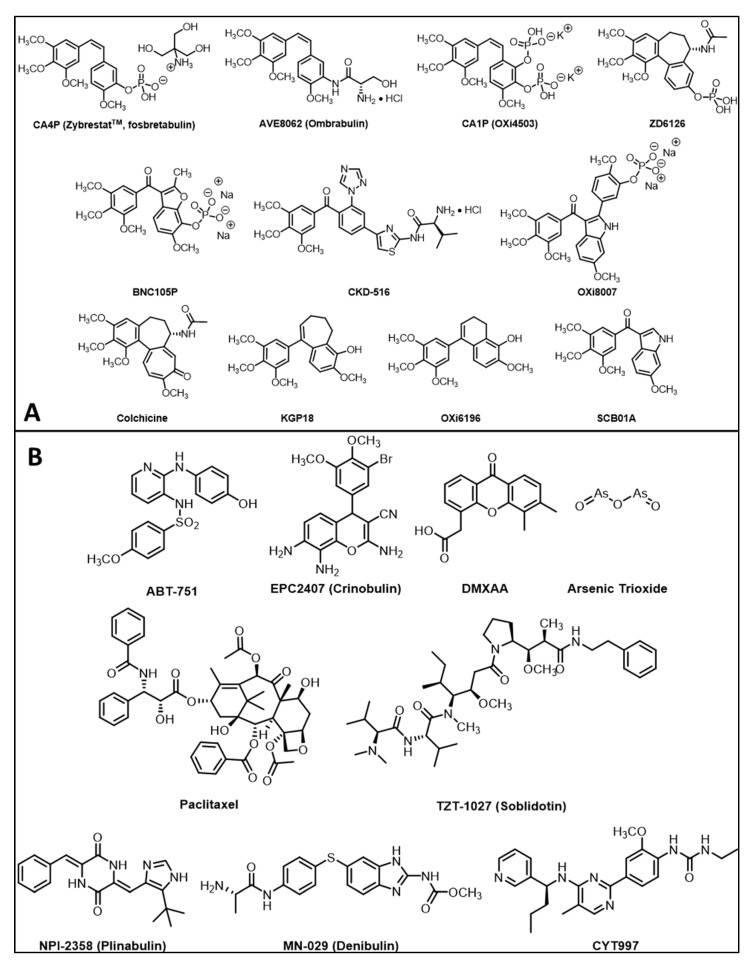
Structures of small-molecule vascular disrupting agents. (**A**) Natural products and combretastatin-inspired molecular structures found to be effective tubulin binding agents causing vascular disruption. Combretastatin A4 [15], AVE8062 [155], combretastatin A1P [14], ZD6126 [156], BNC105P [157], CKD-516 [158], OXi8007 [17], colchicine, KGP18 [144], OXi6916 [159] and SCB01A [160]. (**B**) Diverse molecular structures binding tubulin or causing vascular disruption: ABT-751 [143], EPC2407 [131], DMXAA [138], arsenic trioxide [161], paclitaxel [162], TZT-1027 [163], NPI-235 [124], MN-029 [39] and CYT997 [34]. While paclitaxel is a tubulin binding agent, we found no evidence for acute vascular shutdown (see Figure 6e).

Ultimately, therapeutic efficacy is determined by tumor growth delay and ideally tumor control. However, a crucial intermediate assay is acute vascular disruption. Historically, this was evaluated histologically in excised tumor tissue. Blood vessels themselves can be observed using antibody stains such as anti-CD31 [7,164,165] (Figure 3). However, in the context of vascular disruption it is dynamic changes in flow, perfusion and vascular patency that are critical. These have been measured using perfusion markers such as Hoechst 33342 dye, indocyanine green, DiOC7, colored microspheres or radiolabeled iodoantipyrine (IAP) (^125^I, ^14^C) to reveal extent of perfusion [17,75,157,166,167,168]. Pulse chase approaches with differentially colored dyes, microspheres or lectins allow direct interrogation of changes within specific blood vessels of individual tumors [169,170]. Given the significance of VDAs, there have been several previous reviews [22]. [6,19,25,47,71,171,172], but few have focused on the ability to examine activity non-invasively [5,132,173,174], the emphasis of this current review.

## 2. Imaging Technologies

Imaging provides non-invasive insights into VDA activity in vivo and potentially early predictive biomarkers of therapeutic response. Many non-invasive imaging methods are available [175], and diverse modalities have been applied to evaluate VDAs, including radionuclide approaches, MRI, ultrasound, and more recently photoacoustic and optical (Table 2), as discussed in the following sections. While the focus of this review is on evaluating therapeutic vascular disrupting activity on tumors, it must be remembered that there may also be off-target toxicity [176], which can also be assessed by imaging. Many of these technologies also allow translational imaging in patients, offering the potential for companion studies of efficacy [177].

### 2.1. Bioluminescence Imaging (BLI)

BLI is likely the simplest imaging modality for pre-clinical cancer investigations and has found widespread application to monitor tumor development, growth and metastatic spread, particularly in mice [178,179]. Tumor cells must have been transfected to express luciferase and luciferin substrate is required, though some other enzyme/substrate pairs are becoming available [179]. Instrumentation simply requires a sensitive CCD camera and dark observation chamber and the earliest instruments were built in individual laboratories [89,180]. Sensitivity and ease of use were greatly improved in commercial systems, which became available some 15 years ago and popularity was stimulated by the IVIS systems developed by Xenogen and now produced by Perkin Elmer. There are currently several manufacturers marketing systems with various levels of sensitivity, throughput and additional capabilities. 

In Vivo Optical Imaging Systems are available from: PerkinElmer (Waltham, MA, USA) [181], Spectral Instruments (Tucson, AZ, USA) [182], Scintica (Webster, TX, USA) [183], Sonovol (Durham, NC, USA) [184], Medilume (Montreal, QC, Canada) [185], Vieworks (Anyang-si, Gyeonggi-do, Republic of Korea) [186]. Current systems can typically image five mice simultaneously and in several cases include planar X-ray capability to provide skeletal context. 3D capability has been developed based on multiple angle detection using multiple cameras or mirrors [187,188,189] or depth resolved wavelength dependent spectroscopy [189,190]. Anatomical context is normally provided by overlay on a gray scale surface image of the mouse, but planar X-rays can provide skeletal co-registration, which is particularly relevant for investigating bone metastases [191]. A recent innovation is combination with ultrasound to reveal soft tissue anatomy and vasculature [184]. Mice must be anesthetized and many systems have onboard vaporizers to deliver isoflurane in air or oxygen, though injectable anesthetics such as ketamine/xylazine or pentobarbital have been used. We favor oxygen carrier gas, which promotes survival, during the stress of anesthesia. While imaging is non-invasive, mere anesthetization of animals with a high tumor burden receiving an experimental therapeutic can cause animal loss. BLI is of course limited to observing tissues, which have been transfected to express luciferase. Transfection optimally generates a single high expressing clone, though cells selected for high luciferase expression may no longer fully represent typical heterogeneous tumors. Meanwhile, polyclonal approaches may suffer from differential cellular expression, whereby faster growing clones, which may have lower expression, dominate tumor development in vivo. A typical experimental approach is described in Appendix B. Luciferin is now readily available from many sources and the price has fallen sufficiently that the luciferin substrate required for a typical investigation costs less than $1 per mouse.

The reporter substrate luciferin can be injected directly into a tumor, but luciferin has a remarkable ability to cross membranes including the placenta and blood brain barrier and thus systemic delivery works effectively. Luciferin may be administered intravenously (IV), but this leads to rapid clearance kinetics and is technically challenging for routine use, particularly if multiple sequential doses are to be delivered. Traditional administration was intraperitoneal (IP), but we favor subcutaneous (SC) in the fore-back neck region [89,180,192]. Luciferin rapidly reaches the bloodstream, whereupon it is carried to tumor cells wherever they are located in a mouse and undergoes a light emitting reaction catalyzed by luciferase according to Equation (1).
ATP + *D*-luciferin + O_2_ — ^luciferase^ ⟶ oxyluciferin + AMP + PPi + CO_2_ + light(1)

Signal intensity generally increases for a period of about 5–15 min followed by decline over the next ½ h, though specific kinetics depend on the extent of vascularization and disease site (Figure 3, Figure 4, Figure 5 and Figure 6). Highly reproducible series of images may be observed following repeat administration of luciferin over a period of several hours (Figure 3a), though signal may increase over days due to tumor growth sometimes noticeable within 24 to 48 h (Figure 6a). Recognizing the dynamic variation in signal following administration of luciferin, we favor observing a whole time course, rather than a single time point, and light emission may be compared based on area under the light emission curve, maximum intensity, or intensity at a specific time post administration. Since luciferin must be transported to the tumor, vascular shutdown following administration of a VDA is revealed by reduced light emission (Figure 3, Figure 4, Figure 5 and Figure 6, Appendix A). Indeed, very similar activity was observed in three distinct breast tumors growing orthotopically in the upper mammary fat pad of SCID mice indicating >80% signal reduction within 2 to 4 h following CA4P administration at 120 mg/kg IP (Figure 3 and Figure 4). Reduced perfusion was confirmed using fluorescence microscopy, whereby anti-CD31 staining indicated extensive tumor vasculature, but Hoechst 33342 perfusion dye showed much less accumulation when administered 2 or 4 h after CA4P (Figure 3c).

Optical techniques are subject to attenuation at depth due to absorption and scattering of light, and thus superficial tumors are most easily detected, specifically subcutaneous, or at disease sites near the surface such as mammary fat pad (Figure 3). However, light can penetrate several millimeters and effective signal is observed by BLI in deeper tissues such as the lungs [187,193,194,195], prostate [196,197,198], brain [199,200], pancreas [201,202], liver [203,204], head and neck [114], bone [205,206] and kidney [138,207] in mice. The most common implementation of BLI in oncology is simply to relate signal intensity to tumor burden and indeed several studies have shown that there is a strong correlation for untreated control tumors up to about 2 cm^3^ [89,208]. Beyond this size, absorption of light emitted from deeper tissues by overlaying tumor causes attenuation and a signal intensity plateau [208]. Acute vascular shutdown prevents substrate luciferin from reaching tumor cells, thereby yielding less light and revealing ischemia, as we exploit to observe the effects of VDAs (Figure 3, Figure 4, Figure 5 and Figure 6). 

BLI of VDA activity has most commonly been applied to primary implanted tumors, but investigations have also examined pseudometastases, such as breast tumor colonization of the lungs, following tail vein injection of MDA-MB 231-luc cells (Figure 4b). Some primary tumors effectively yield spontaneous metastases, which are readily observed, although the primary tumor may need to be masked for effective imaging due to relative signal intensities (Figure 4c,d). Dynamic BLI was initially applied to known VDAs to establish the technique, e.g., CA4P [209] (Figure 3 and Figure 4), CA1P [114], DMXAA [134,210], BPR0L075 [129], and ATO [161]. It has also been validated by comparing changes in BLI signal against alternative technologies such as MRI [209], ultrasound [161], photoacoustics [92] and histology (Figure 3), each of which has shown effectively correlated data. This provided confidence to apply dynamic BLI to new agents such as the indole OXi8007 [17,93], and efficacy is shown against orthotopic breast and kidney tumors in Figure 5. BLI is particularly effective at demonstrating dose response for new potential VDAs [17,66,74,145,146,147,211], as exemplified for a novel amino benzosuberene analog (KGP321) of KGP18 (Figure 2A and Figure 6). The combretastatins CA4P and CA1P and indole analog OXi8007 and benzosuberene analog KGP265 are administered as water soluble phosphate prodrugs, which are readily dephosphorylated releasing the active molecules, which bind tubulin and cause microtubule disaggregation yielding vascular collapse and ischemia, as evident from reduced BLI signal. Meanwhile, the microtubule stabilizing therapeutic paclitaxel caused no signal loss over 48 h (Figure 6e) matching a reported lack of change in BLI signal from 4T1-luc breast tumors growing as pseudometastases in the mouse lung after IV injection of cells [212]. 

Dynamic BLI is very effective for initial evaluation of potential VDAs, providing high throughput results with relative ease and low cost. There are distinct limitations, though many are readily overcome: (i) requires transfected cells to express luciferase effectively (many stably transfected cell lines are now available); (ii) requires administration of luciferin substrate (currently readily available, cheap and non-toxic); (iii) requires optical imaging system (typically cheapest of all available modalities); (iv) typically limited to mice due to light scattering by tissues, although BLI has been reported in rats including prostate [198], brain [178,213] and lung tumors [214]. Other more sophisticated technologies provide alternative imaging approaches. 

### 2.2. Magnetic Resonance Imaging (MRI)

Historically, MRI was the most commonly used modality for non-invasive investigations of VDA activity. MRI not only provides detailed anatomical images and high spatial resolution to reveal tumor location and heterogeneity, but can also provide insights into pathophysiology and pharmacodynamics in response to interventions. Pre-clinical MRI uses a high field magnet (typically, 3 to 11.7 Tesla available from two primary manufacturers for pre-clinical investigations: Bruker (Billerica, MA, USA) [215] and MR Solutions (Guildford, Surrey, UK) [216]) to polarize the nuclear magnetic spins and radio frequency stimulation to excite signals, most commonly the tissue water. Applied magnetic field gradients provide spatial resolution and signal relaxation produces contrast. Typical images in living mice have submillimeter in-plane resolution and millimeter slice thickness and may be presented as two-dimensional tomographic or three-dimensional volume images. MRI is particularly versatile in terms of pulse sequences, whereby combinations of excitation pulses and gradients can interrogate specific aspects of physiology, such as perfusion, flow, necrosis, and oxygenation [177]. Administration of paramagnetic contrast agents reveals flow, perfusion and vascular leakage, as applied in pre-clinical studies to develop many VDAs and subsequent clinical investigations (Table 1 and Table 3). 

The most common application regarding VDA activity examines changes in tissue contrast accompanying intravenous infusion of a paramagnetic contrast agent such as gadolinium-DTPA (Gd-DTPA, Magnevist), referred to as dynamic contrast enhanced (DCE) MRI. Spin lattice relaxation rate (R_1_ = 1/T_1_) is directly related to the concentration of small molecule contrast agents. Following IV infusion of a contrast agent, the inflow, perfusion and accumulation due to leakage and clearance are observed readily revealing changes following administration of VDAs. Usually the initial area under the contrast curve (IAUC) is used to examine perfusion, often characterized by the signal amplitude (%ΔSI) and time to reach maximum (TTM) (Figure 7). More sophisticated analysis can reveal perfusion kinetics; to provide rigor with quantitative approaches the arterial input function (AIF) is required or a reference tissue model may be applied [217] (Figure 7). The reference tissue model compares the contrast agent curves in a tissue of interest (tumor) to that of a reference region (muscle). A typical experimental approach is described in Appendix C. Using reported values for the volume transfer constant of muscle, K^trans^, M, (0.1 min^−1^) and the extravascular-extracellular volume fraction, v_e_, M, (0.1) in the muscle [217,218], it is possible to extract the K^trans^ and v_e_ values for the tumor without knowledge of the AIF. The contrast concentration curve, C(t), in tumor tissue is then given by [217]:C(t) = R*C^M^(t) + R*[(K^transM^/v_e_^M^) − (K^trans^/ v_e_)]*_0_∫^t^ C^M^(t′) * exp (−(K^trans^/v_e_)*(t − t′)dt(2)
where R = (K^trans^/ K^transM^) and C^M^(t) is the contrast agent curve in muscle tissue. K^trans^/v_e_ is often referred to as k_ep_. 

Widespread DCE investigations were used in the development of the combretastatins (CA4P and CA1P) as well as DMXAA and ZD6126) [5,27,37,96,97,100,102,103,104,106,117,124,125,126,127,128,131,132,135,136,137,138,219,220,221] and continue to be popular in pre-clinical development (Table 3), since they may also be implemented readily in clinical trials (Table 1). Examples of parametric images derived from DCE-MRI are presented in Figure 7, where tumor heterogeneity is apparent at baseline, as well as in response to the novel vascular disrupting agent OXi6197, the phosphate prodrug of OXi6196 (Figure 2A) a dihydronaphthalene analog of combretastatin. It is apparent that the initial area under the curve is much lower 24 h after VDA, while the TTM is much greater in many regions. The extracellular-extra vascular volume (v_e_) is considerably greater indicating enhanced vascular permeability. Others have used polymeric or particulate contrast agents (e.g., ultra small paramagnetic iron oxide: USPIO), which are retained in the vasculature for longer periods giving a sustained indication of vessel caliber and induced vascular leakage [106,127,128].

**Table 3 molecules-26-02551-t003:** Imaging VDA activity using MRI.

MRI ^a^	VDA	Tumor Type	References
Perfusion/flow/vascular permeability DCE; DSC	CA4P, DMXAA, EPC2407, ZD6126, CKD-516	Liver, colorectal, pancreatic, breast, glioma, prostate, carcinosarcoma, VX2, kidney, H&N	[5,27,37,96,97,100,102,103,104,105,106,117,124,125,126,127,128,131,132,135,136,137,138,219,220,221]
Diffusion DWI/IVIM	CA4P, CKD-516	Liver, rhabdomyosarcoma, lung, VX2	[98,102,104,117,219,222]
ASL	CA1P	Colorectal	[116]
BOLD, OE-MRI, ^19^F oximetry	ZD6126, CA4P, OXi8007	Breast, bladder	[5,93,94,95,223]
CEST	TNF-α	SC colon tumors	[224]
pH	CA4P, DMXAA, ZD6126	Breast	[91]
HP-pyruvate	CA1P, CA4P	Lymphoma, Breast	[105,225]

**^a^** DSC: dynamic susceptibility contrast; IVIM: intra voxel incoherent motion; OE: oxygen-enhanced; CEST: chemical exchange saturation transfer; HP hyperpolarized.

While MRI was incorporated into many clinical trials regarding anti-angiogenesis agents, there have been far fewer trials with respect to vascular disrupting agents. In principle, assessment of VDAs should be particularly easy since response is generally acute and substantial, however in terms of DCE-MRI, various questions remain open: which parameter is most relevant (e.g., changes in v_e_ or K^trans^); when should imaging be performed; what threshold serves as a useful predictive biomarker [24]?

DCE-MRI has been used for many years and implementation and analysis are quite straightforward and readily translated to human studies. A downside to DCE-MRI is the need for the paramagnetic contrast agent; in rodents, this adds to the complexity by requiring effective IV infusion, while in human subjects there is recent concern that the gadolinium ions may cause kidney damage due to nephrogenic systemic fibrosis (NSF), or from as yet unknown problems due to deposition and long-term retention in the brain [226,227]. It appears that cyclic paramagnetic contrast agents have fewer potential issues, but contrast agent administration is now avoided in patients when possible. As such, various other MRI methods have been developed. Notably, arterial spin labeling allows non-invasive observation of blood flow based on physical spin tagging alone [116]. It has been effectively applied to colorectal cancer in mice revealing ischemia, but the method does require adequate through-plane signal motion to reveal flow. Acute vascular shutdown is the most obvious early effect revealed by MRI. Later effects may also be examined such as using diffusion-weighted imaging (DWI) to observe changes in cell structure, whereby diffusion is restricted in well-structured tissue, but becomes more mobile with necrosis, which occurs extensively within 24 h after VDA [98,102,104,117,219,222]. MR Elastography indicated reduction in tumor viscoelasticity 24 h after ZD6126 at which time no significant change in tumor apparent diffusion coefficient (ADC) was observed [228].

Hypoxia may be observed using oxygen-sensitive MRI including Blood Oxygen Level Dependent (BOLD) and Tissue Oxygen Level Dependent (TOLD) contrast [229,230,231,232,233]. In principle, vascular disruption causes ischemia and yields hypoxiation. Deoxyhemoglobin (Hb) is paramagnetic leading to accelerated dephasing in transverse relaxation and increased R_2_* (1/T_2_*) related to [Hb] [126]. Indeed, Robinson et al. found a dose response increase in R_2_* in both prolactinomas and RIF-1 fibrosarcomas following ZD6126 administration [126]. However, vascular collapse could lead to exclusion of blood from the tumor generating an opposite effect. Thomas et al. examined the BOLD response to a carbogen breathing challenge before and after administration of CA4P (100 mg/kg) to rat bladder tumor bearing mice and found much reduced response within 35 min [94]. The spin lattice relaxation rate, R_1_, is directly sensitive to pO_2_. One day after administering OXi6197 to a rat bearing a subcutaneous A549 lung tumor in the leg, the tumor R_2_* and R_1_ showed little change under baseline air breathing conditions, but now there was much less response to an oxygen gas breathing challenge (much reduced ΔR_1_ and ΔR_2_*) indicating impaired perfusion and implying hypoxiation (Figure 7B). Changes in R_1_ and R_2_* appear to be particularly effective at identifying tumor perfusion and lack of response is associated with hypoxia [231].
Hb + O_2_ ⟶ HbO_2_
(3)

However, high concentrations of Hb, e.g., accompanying hemorrhage can also accelerate R_1_, giving potentially anomalous results [233,234]. The signal changes observed using TOLD in response to a hyperoxic gas breathing challenge are much smaller than seen in DCE following Gd-contrast and thus TOLD must be performed first, as in Figure 7, or separate cohorts of animals used [95].

Hypoxiation is expected to enhance metabolic dependence on glycolysis, but the ratio of pyruvate to lactate was unchanged based on HP (hyperpolarized)-MRI [105], and others reported decreased intrinsic lactate 24 h after ZD6126 [235]. ^31^P NMR indicated loss of high energy phosphate metabolites (ATP and PCr) and increased lactate in HT29 tumors 6 h following DMXAA [236]. Hypoxiation has been definitively examined using ^19^F MRI of the reporter molecule hexafluorobenzene (HFB) following intra tumoral injection and administration of OXi8007 IP [93]. Meanwhile, Diepart et al. [80] found increased pO_2_ based on ^19^F MRI of HFB in TLT tumors following administration of low dose ATO, which acted as a mitochondrial inhibitor, as opposed to showing the vascular disrupting activity observed at high doses [161].

MRI is particularly effective for observing the effects of VDA administration, but does require substantial expertise, magnets and associated hardware. In pre-clinical studies, it provides effective characterization of tumor heterogeneity, but suffers from low throughput and need for sophisticated image acquisition and data analysis, thereby reducing efficiency. The great strength is that observation may be made in rodents and reproduced in humans. MRI may not be feasible if the subject has metal implants or a pacemaker, or is particularly obese or severely claustrophobic.

### 2.3. Photoacoustics-Multispectral Optoacoustic Tomography (MSOT)

Photoacoustics has recently become feasible in many laboratories with the availability of commercial small animal imaging systems, as well as experimental devices suitable for patients in clinical trials (iThera (Munich, Germany) [237] and Fuji Visual Sonics (Toronto, ON, Canada) [238]. Notably, the iThera MSOT and Fuji Visual Sonics LAZR-X systems readily provide rapid high spatial resolution images (approaching 120 µm in plane) of oxy- and deoxyhemoglobin. Single wavelength images may be acquired in 100 ms each, but the need for signal averaging and acquisition of several wavelengths to allow spectral unmixing for precise quantitation, typically provides temporal resolution of 1 to 20 s. While the concept of photoacoustics was described over 100 years ago, it represents the newest modality available for probing dynamic vascular disrupting activity in vivo. Pulsed light at a specific wavelength is selectively absorbed by chromophores depending on concentration yielding local thermoelastic expansion and generating shock waves, which are detected using ultrasound transducers to provide spatial resolution. Oxy- and deoxyhemoglobin are both abundant and strongly absorbing, with well-defined separate spectra allowing vascular oxygen saturation to be estimated. 

The iThera MSOT operates through excitation of a tissue plane (~1 mm thick) using 10 fiber optic excitation bundles and an array of 128, 256 or 512 acoustic transducers placed toroidally around the subject providing a bore size of 4 cm, which is ideal for investigating mice. Meanwhile, the LAZR excites and detects from a single direction, allowing the interrogation of larger subjects, but providing less uniform excitation and detection. The LAZR-X has the advantage of incorporating ultrasound excitation, which provides enhanced anatomical imaging and Doppler flow detection. This is also available on the iThera Acuity system, which is designed for larger animals and human investigations. Several studies have reported dynamic vascular response of tumors in mice or rats to vascular disruption caused by CA4P, CA1P and EPC2407 [92,118,119,130,239,240]. In most cases, changes in [HbO_2_] and [Hb] are examined, thereby also revealing vascular oxygen saturation (sO_2_). Ron et al. explored natural cycling hypoxia in human breast tumor xenografts in mice [241]. Tomaszewski et al. showed rigorously that the response to an oxygen gas breathing challenge provides a better picture of tumor oxygenation than a static baseline measure of sO_2_ alone [240]. Additionally, blood flow and perfusion may be revealed by DCE-MSOT following IV infusion of a strongly absorbing contrast agent such as ICG, Genhance or gold nanoparticles [240,242,243].

Rich & Seshadri compared oxygen-sensitive MRI and photoacoustics to cross validate the observations [130], while Tomaszewski et al. compared oxygen-enhanced-MSOT and DCE-MSOT [240]. In other studies, immunohistochemistry was used to validate vascular perfusion and hypoxiation [92,240]. The commercial systems can observe large blood vessels and general oxygenation of the capillary bed (individual capillaries do not need to be resolved). Meanwhile, much higher resolution photoacoustic microcopy has been used to examine the vascular tree and show oxygenation and hypoxiation in individual blood vessels following administration of VDA [118].

Matching recent reports, we used an InVision 256-TF small animal imaging system to reveal spatial heterogeneity of tumor vasculature (Figure 8, Figure 9 and Figure 10). We have applied three different paradigms to examine vascular disruption. Firstly, dynamic observation revealed changes in tumor hemoglobin oxygen-saturation response (ΔsO_2_) accompanying an O_2_-breathing challenge (Figure 8, Figure 9 and Figure 10). At baseline, the tumor center of MDA-MB-231 tumors showed little response, but the tumor periphery showed a significant change, as also seen in the muscle surrounding the spine (Figure 8). Following administration of VDA the response to oxygen gas breathing challenge was much smaller as seen in the orthotopic MDA-MB-231 breast tumor (Figure 8), a syngeneic orthotopic RENCA kidney tumor (Figure 9) and a subcutaneous human A549 lung tumor xenograft in the leg of a rat (Figure 10). The response to an oxygen gas breathing challenge appears far more effective at revealing vascular patency and hypoxia compared with baseline measurements alone [244,245]. Secondly, progressive hypoxiation may be observed by continuous imaging following administration of VDA (Figure 8 and Figure 9). Histograms of vascular oxygenation emphasized distinct hypoxiation (Figure 8 and Figure 9). Response to oxygen gas challenge and progressive hypoxiation are apparent in the video (See Appendix A). Thirdly, DCE-MSOT based on the pharmacokinetic distribution of a contrast agent such as ICG or Genhance reveals changes in perfusion [242,243]. MSOT is often applied to a single imaging plane through a tumor, but acquisition of multiple slices can readily provide 3D information. A typical experimental approach is described in Appendix D.

Similar to the breast tumors in Figure 8, vascular disruption has also been observed based on an oxygen gas breathing challenge at successive time points in orthotopic syngeneic RENCA tumors (Figure 9). In addition to detecting vascular shutdown based on an oxygen gas breathing challenge DCE-MSOT showed much reduced perfusion based on Genhance, while the contralateral kidney was largely unaffected by VDA (Figure 9b). MSOT is particularly well suited for investigations of the kidney and renal cell carcinoma (RCC) because of the extensive vasculature. The RENCA tumors also expressed luciferase and could have been assessed using BLI, but such transfection is not feasible in primary tumors (PDX and GEM models). We have also observed hypoxiation in human XP373 explant tumors following administration of OXi8007 IP (Figure 9). Hong et al. used MSOT to show that application of DMXAA could enhance the local trapping of gold nanoparticles in CT26 tumors thereby increasing the effectiveness of photothermal excitation and tumor growth delay [133]. Photoacoustic imaging revealed the uptake of gold nanoparticles and BLI was used to assess tumor progress following therapy. Liu et al. used MSOT to examine the activity of a novel poly(L-glutamic acid)-CA conjugate (PLG-CA4) designed to restrict transport and enhance accumulation in tumors [154]. In addition to evaluating changes in vascular oxygenation they added an IR820 fluorochrome allowing direct visualization of the PLG-CA4 itself. While most MSOT studies of VDA activity have been performed in tumor bearing mice, we have also observed vascular disruption in lung tumor xenografts growing in the leg of a rat (Figure 10). The A549 lung tumor has extremely sparse vasculature [246], but this is initially highly responsive to an oxygen gas breathing challenge. Two days after administration of the novel combretastatin analog OXi6197, the remaining vasculature showed minimal response, matching the MRI performed on the same tumors (Figure 7).

While spectral unmixing directly indicates relative concentrations of oxy- and de- oxyhemoglobin, spectral coloring can perturb effective quantitation at depth due to differential light absorption by overlying tissues. Recent studies have sought to compensate for spectral coloring [242]. Rapid image acquisition reveals motion [247], which could compromise effective spectral unmixing indicating a need for image co-registration. Acquisition of individual spectral frames can avoid averaging misregistered images, potentially allowing corrections or deletion of offset images. Additionally, appropriate filtering algorithms can enhance signal to noise and contrast, though they may also introduce damping into transitions [248].

It should be noted that oxygen breathing challenge (the simplest theranostic) is recognized as optimal for assessing vascular patency, as we have often used for MRI [231,249,250] and we and others now apply to MSOT [92,240,244,245]. Exogenous vascular contrast agents can also be used (e.g., ICG or Genhance) [240,242,243] and could be particularly relevant for sparse vasculature where there is little hemoglobin signal (e.g., Figure 10).

MSOT directly reveals relative spatial and temporal variations of oxy- and deoxyhemoglobin concentrations without the need for exogenous reporter molecules or cell transfection with reporter genes [245,251,252]. While investigators often present sO_2_, it should be noted that values of [HbO_2_] and [Hb] are not strictly quantitative and thus vascular oxygen saturation should be described as sO_2_^MSOT^. Limited anatomical contrast is also observed (Figure 8 and Figure 9). MSOT is however, a complicated procedure requiring considerable practice to achieve effective images and requiring sophisticated analysis. While all tissue may be observed in a mouse, the method becomes ineffective in melanin rich dark skinned mice. Rigorous hair removal essential, effective optical and acoustic contact are required using acoustic gel and it is important to ensure there are no bubbles. Some systems require mice to be immersed under water with breathing via a snorkel. Rapid image acquisition means that respiratory and cardiac motion may be observed and potentially co-registration should be applied to minimize motion artifacts. Animals are imaged individually. Photoacoustics may be enhanced with the development of new contrast agents, an area of active investigation [252,253].

### 2.4. Other Modalities

Ultrasound imaging (US) applies sound waves to a subject and receives echoes, which provide anatomical structures with exceedingly high spatial and temporal resolution. Blood flow and vasculature may be observed using color- or power-Doppler methods. These approaches have been applied to assess VDA activity in various species ranging from mouse to rat, rabbit, dog and human subjects [5,74,91,109,110,111,112,161]. Use of high frequency sound waves provides finer spatial resolution, but limits depth of signal penetration. Sound waves are also subject to severe scattering by bones and do not transmit through air. US can be entirely non-invasive, but Doppler methods do require sufficient flow to provide an observable signal. Examples of color-Doppler imaging of VDA activity are presented in Figure 11, for subcutaneous human tumor xenografts. The lung tumor showed very sparse vasculature, while much stronger flow was seen in the prostate tumor, with each showing substantial diminution after about 1.5 h. A typical experimental approach is described in Appendix E. Observation of low flow regions may be enhanced using microbubble contrast [5], as exploited extensively by Abma et al. in canine subjects [109,254]. Alhasan et al. compared power Doppler with dynamic BLI showing consistent diminution in flow following administration of ATO based on both methods [161]. Meanwhile, fluorescent signal from constitutively expressed mCherry in transfected tumor cells showed no changes over 24 h following ATO.

Positron Emission Tomography (PET) uses coincidence detection of gamma rays following annihilation of a positron to determine the site of radioactive decay in a subject following administration of a positron-emitting isotope. Typical isotopes include ^15^O, ^11^C and ^13^N for labeled gases and water. Meanwhile, ^18^F FDG is now routinely used to assess metastatic spread based on the metabolic hyperactivity of tumors. PET is exceptionally sensitive allowing the use of very low concentrations of tracer, but spatial resolution is typically much poorer than MRI and temporal resolution is slow. The biggest obstacle to routine use is the need to safely handle radioactivity and the rapid decay of isotopes that have typically been used to assess flow (T_1/2_
^15^O = 122 s). There has been comparison of FDG PET and ^13^C hyperpolarized NMR of pyruvate [105]. Uptake of ^18^F-FDG in subcutaneous U251 glioblastoma xenografts was reduced following administration of DMXAA, likely due to both vascular disruption impeding delivery of FDG and also cell death after 24 h [141].

Single photon emission computed tomography (SPECT). Other radioisotope approaches have explored SPECT/CT of ^131^I-Hoechst 33258 (^131^I-H33258) in W256 tumor-bearing rats as an early predictive biomarker of tumor response to CA4P based on its avidity for necrotic tissue [113]. Planar scintigraphy was used to explore hypoxia in subcutaneous RIF-1 tumors following DMXAA or CA4P [255]. In each case ^99m^Tc-labeled HL-91 (Prognox) and VDA were administered simultaneously and mice imaged 3 h later revealing increased uptake of ^99m^Tc and hence increased hypoxia. Studies additionally applied ^86^RbCl to assess perfusion, though this required sacrifice and correction for residual radioactivity from the ^99m^Tc.

Optical imaging. Tissue perfusion may be effectively observed based on fluorescent-tagged substrates, though this is typically limited to superficial tissues. Dynamic contrast enhanced fluorescent imaging (DyCE FLI) has been applied to examine perfusion of subcutaneous tumors implanted in the backs of mice and reduced flow was observed following administration of CA4P [256]. Laser Doppler flowmetry can directly reveal flow, though signal is limited to very superficial tissues. Optical microscopy has been applied to tumors in superficial dorsal window chamber modes where vascular disruption could be assessed at high resolution, showing shutdown or individual vessels, as well as induced hyperactive permeability and leakage of contrast agents [257]. Hyperspectral imaging revealed changes in vascular oxygen saturation.

### 2.5. Optimizing Combination Therapy

As mentioned in the Introduction, effective treatment with VDAs will likely require combination therapy, specifically to overcome the peripheral surviving tumor tissue ring observed in most studies [11,41,56,258]. Imaging should be particularly effective in facilitating optimal combination based on timing and extent of acute vascular changes. Notably, enhanced permeability may promote delivery of additional drugs, though ischemia may limit access. Consequent hypoxia is expected to reduce the efficacy of radiation therapy, though could promote hypoxia activated pro-drugs [259]. 

Building on earlier DCE MRI evaluations of tumor perfusion following VDA treatment [106], Fruytier et al. established that delivery of gemcitabine was diminished in TLT hepatocarcinomas growing intramuscularly in mice 2 h after CA4P. They elegantly showed changes in vascular perfusion and permeability based on DCE-MRI, as well as assessing uptake and metabolic conversion of gemcitabine using ^19^F NMR spectroscopy [107]. Such an approach could allow effective determination of optimal timing of combined therapy by establishing both the pharmacokinetics and uptake of a particular therapeutic agent, as well as the pharmacodynamic vascular perturbation caused by the VDA.

Folaron et al. examined “vascular priming” to enhance the efficacy of several common chemotherapeutic drugs through combination with DMXAA. They specifically applied dynamic BLI to examine tumor growth and DCE-MRI to evaluate changes in tumor perfusion and permeability in relation to the efficacy of irinotecan, docetaxel, and doxorubicin [134]. Intriguingly, they found increased BLI signal 1 h after DMXAA suggesting enhanced delivery of luciferin substrate, but significantly diminished signal at 24 h. consistent with vascular collapse, as confirmed by DCE-MRI.

Delivery of chemotherapeutic agents can be enhanced through active targeting and encapsulation in nanoparticles to improve local retention. Sun et al. designed a “cooperative polymeric platform” for tumor-targeted drug delivery. Recognizing that the peptide GNQEQVSPLTLLKXC (A15) is a substrate of activated blood coagulation factor XIII (a transglutaminase), they created A15 peptide-decorated poly(L-glutamic acid)-cisplatin conjugates as coagulation-targeted nanoparticles [140]. They then exploited the VDA DMXAA to induce hemorrhage in tumors yielding “a unique coagulation environment”. Using MSOT they were able to convincingly show elevated uptake of such fluorescently (NIR830)-labeled NPs after administration of DMXAA and ultimately demonstrate enhanced tumor growth delay in mice. 

Zhao et al. showed distinct hypoxiation of 13762NF rat breast tumors within 30 min of 30 mg/kg CA4P using near infrared spectroscopy and additionally found that the tumors became essentially unresponsive to an oxygen gas breathing challenge at 2 h, although some response was restored after 24 h [260]. This coincided with greatest tumor growth delay being achieved when tumors were irradiated (5 Gy), while rats breathed oxygen 24 h after CA4P, whereas other sequences of treatment were less effective. Diepart et al. used ^19^F MRI to determine that tumor pO_2_ increased transiently for about 2 h after administration of ATO (5 mg/kg) to mice [80]. This suggested tumor irradiation at 90 min following ATO would be particularly effective, as indeed observed. 

VDA induced hypoxia has also been explored to promote activation of bioreductively-activated prodrugs [148,261]. Notably, Shen et al. developed a paradigm for modulation of host immunological responses during cancer treatment by exploiting CA4 and the immune modulator Imiquimod (IMQ) [261]. Noting that TIE2+ tumor-associated macrophages (MΦ) and endothelial progenitor cells have been reported to infiltrate tumors after treatment with CA4P, thereby promoting tumor angiogenesis, [262], it was reasoned that the immune modulator IMQ could potentially convert immature plasmacytoid dendritic cells (pDCs) into their active form, leading to the robust infiltration and priming of natural killer cells and cytotoxic T-lymphocytes in treated tumors. To seek tumor specificity a bioreducible prodrug hs-IMQ was prepared and co-administered with CA4 in poly(L-glutamic acid)-graft-methoxy poly(ethylene glycol nanoparticles (NPs). The NP depots ensured longer-term delivery/release of CA4, which induced additional tumor hypoxia promoting nitroreductase activity and IMQ release. Therapeutic efficacy in 4T1 breast tumors in mice was enhanced. Pimonidazole was also administered to reveal increased hypoxia in tissue slices commensurate with release of IMQ. In vivo imaging was not applied, but this study represents an ideal opportunity for applying ^18^F-miso PET, oxygen-sensitive MRI or MSOT to examine the dynamic evolution of hypoxia non-invasively.

## 3. Discussion

Diverse imaging modalities are available to provide non-invasive insights into the mode of action of vascular disrupting agents. Technologies offer various levels of resolution, sophistication, complexity and cost in terms of implementation as summarized in Table 4. The simplest observations are sensitive to flow and perfusion and readily reveal acute changes indicative of vascular disruption. Dose response may be determined efficiently since each tumor serves as its own control. The more sophisticated methods additionally reveal tumor heterogeneity and heterogeneity of response. Beyond flow and ischemia, other technologies can reveal hypoxiation, necrosis and more subtle pathophysiology and metabolism. Imaging has become increasingly routine in small animal preclinical research and is increasingly encouraged in companion studies to optimize clinical trials. Ultimately, assessment of the pharmacokinetics of drug delivery together with the pharmacodynamics of tumor pathophysiological response should enable more effective personalized medicine tailored to optimizing therapeutic efficacy based on predictive/prognostic imaging biomarkers.

## 4. Materials and Methods

The results presented in Figure 1, Figure 2, Figure 3, Figure 4, Figure 5, Figure 6, Figure 7, Figure 8, Figure 9, Figure 10 and Figure 11 are all unpublished though in many cases closely analogous to investigations presented previously. All studies were approved by the UTSW Institutional Animal Care and Use Committee and performed in line with State and Federal guidelines. Specific details are presented in the text and figure legends and general description of the technologies in the Appendix for each technology.

## 5. Patents

Kevin G. Pinney, Haichan Niu, Depoprosad Mondal, “Benzosuberene Analogues and Related Compounds with Activity as Anticancer Agents” (United States Patent: US 10,807,932 B2), issued 20 October 2020.Kevin G. Pinney and Madhavi Sriram, “Combretastatin Analogs with Tubulin Binding Activity” (United States Patent: US 8,394,859), issued 12 March 2013.David J. Chaplin, Klaus Edvardsen, Kevin G. Pinney, Joseph Prezioso, Mark Wood, “Compositions and Methods with Enhanced Therapeutic Activity” (United States Patent: US 8,198,302), issued 12 June 2012.Kevin G. Pinney, Feng Wang, Maria Del Pilar Mejia, “Indole-Containing and Combretastatin-Related Anti-Mitotic and Anti-Tubulin Polymerization Agents” (EP 1 214 298 B1), issued 30 May 2012.

## Figures and Tables

**Figure 1 molecules-26-02551-f001:**
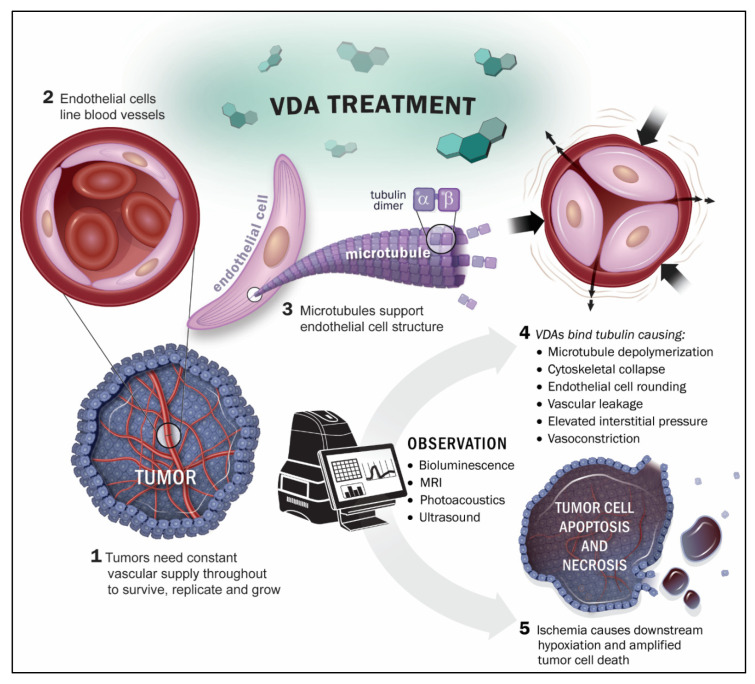
Imaging the action of VDAs on tumor-associated blood vessels.

**Figure 3 molecules-26-02551-f003:**
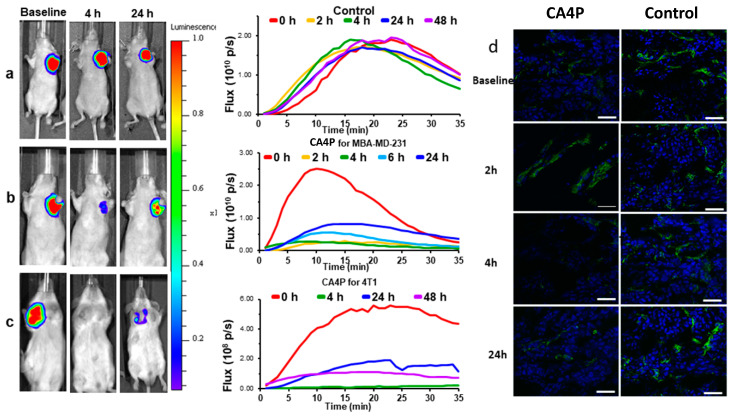
Efficacy of acute VDA activity revealed using dynamic BLI in orthotopic breast tumors. Left: BLI signal intensity images overlaid as heat maps on gray scale photographs of mice at about 10 min after administration of luciferin at selected time points following administration of (**a**) saline or (**b**) CA4P (120 mg/kg IP) to MDA-MB-231-luc xenograft tumor bearing nude mice. See also Appendix A; (**c**) CA4P to syngeneic 4T1-luc tumor in BALB/C mouse. All intensity maps have same heat scale. Center: corresponding BLI intensity curves for the respective individual mice at left showing differential variation over a period of 35 min following administration of luciferin at baseline (red), 2 h post (orange), 4 h (green) and 24 h (blue), 48 h (purple). (**d**) Tumor sections from four tumors showing vascular extent based on CD31 stain (green) and perfusion marker Hoechst 33342 (blue) at different times following treatment with CA4P. Severely diminished perfusion was seen at 4 h, while controls tumors showed highly consistent extensive perfusion as seen in right hand column. Scale bar: 50 µm.

**Figure 4 molecules-26-02551-f004:**
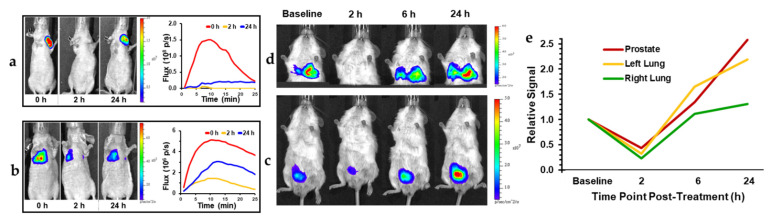
CA4P response for BrCa and PCa with implanted spontaneous lung metastases. (**a**) Orthotopic MCF-7-luc human breast cancer tumor in upper mammary fat pad response to CA4P treatment (120 mg/kg IP) assessed by BLI at 0, 2, and 24 h after treatment. BLI intensity images overlaid on photos of mice, (**b**) Equivalent results for MCF-7-luc lung colonization tumor model generated by IV injection of tumor cells. Corresponding dynamic time course intensity traces are shown for each time following administration of luciferin. At 2 h, there was about 80% reduced light emission corresponding to vascular shutdown in mammary fat pad tumor, but less loss (60%) in the lungs. Similarly, orthotopic PC3-luc tumor (**c**) and its spontaneous lung metastases (**d**) in SCID mouse at baseline, 2, 6 and 24 h following CA4P. (**e**) The PC3 tumors appeared to be somewhat more resistant to VDA activity with less signal loss.

**Figure 5 molecules-26-02551-f005:**
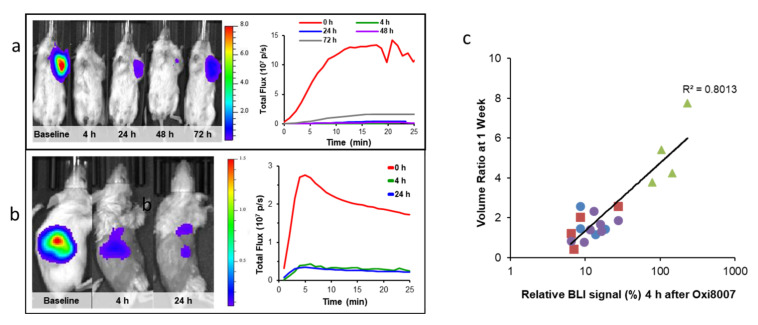
Demonstrating efficacy of new VDA in tumors growing in mice. (**a**) MDA-MB-231-luc breast tumor response to OXi8007 treatment (350 mg/kg IP) assessed by BLI at 4, 24, 48 and 72 h after treatment. BLI intensity images overlaid on photos of mice. Corresponding signal intensity curves at right; (**b**) similarly, orthotopic RENCA-luc kidney tumor bearing mouse at baseline, 4 and 24 h. Dynamic time course intensity traces are shown for each time following administration of luciferin. At 4 and 24 h there was an approximately 80% reduction in light emission corresponding to vascular shutdown and matching the MSOT observation in Figure 10; (**c**) BLI as predictive imaging biomarker. For groups of MDA-MB-231-luc tumors (control, low and high dose OXi8007 as well as combination with carboplatin) there was a strong correlation between the signal shutdown at 4 h after administering VDA and the tumor growth over the next 7 days (R^2^ > 0.8).

**Figure 6 molecules-26-02551-f006:**
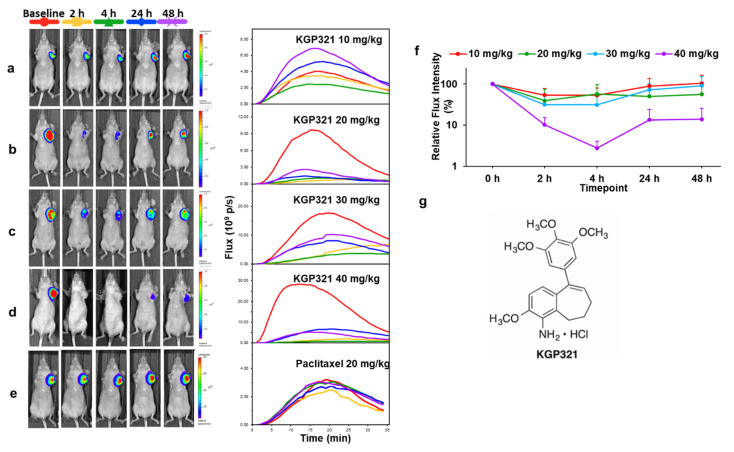
Dose response of MDA-MB-231-luc MFP tumors to KGP321. Left: BLI signal intensity images overlaid as heat maps on gray scale photographs of nude mice at 16 min following administration of luciferin at selected time points following administration of KGP321. Center: BLI intensity curves for the respective individual mice at left show differential variation over a period of 35 min following administration of luciferin at baseline (red), 2 h post (orange), 4 h (green) and 24 h (blue), 48 h (purple). (**a**) 10 mg/kg (*n* = 4); (**b**) 20 mg/kg (*n* = 3); (**c**) 30 mg/kg (*n* = 3); (**d**) 40 mg/kg (*n* = 3); (**e**) Paclitaxel (20 mg/kg) showing no acute changes in BLI signal. (**f**) BLI relative intensity change curves showing variation over a period of 48 h following administration of different doses of KGP321. Line colors 10 mg/kg (red); 20 mg/kg (green); 30 mg/kg blue) and 40 mg/kg (purple). ANOVA based on Fisher’s PLSD indicated that over the 72 h-time course all doses 10–40 mg/kg gave significantly different light emission from vehicle alone (*p* < 0.005), but there was no significant difference between the doses. At 30 and 40 mg/kg there was significantly less light emission at 2 and 4 h compared with baseline (*p* < 0.05) and this continued up to 72 h for 40 mg/kg (*p* < 0.0001). (**g**) Structure of KGP321, a novel aminobenzosuberene-based VDA.

**Figure 7 molecules-26-02551-f007:**
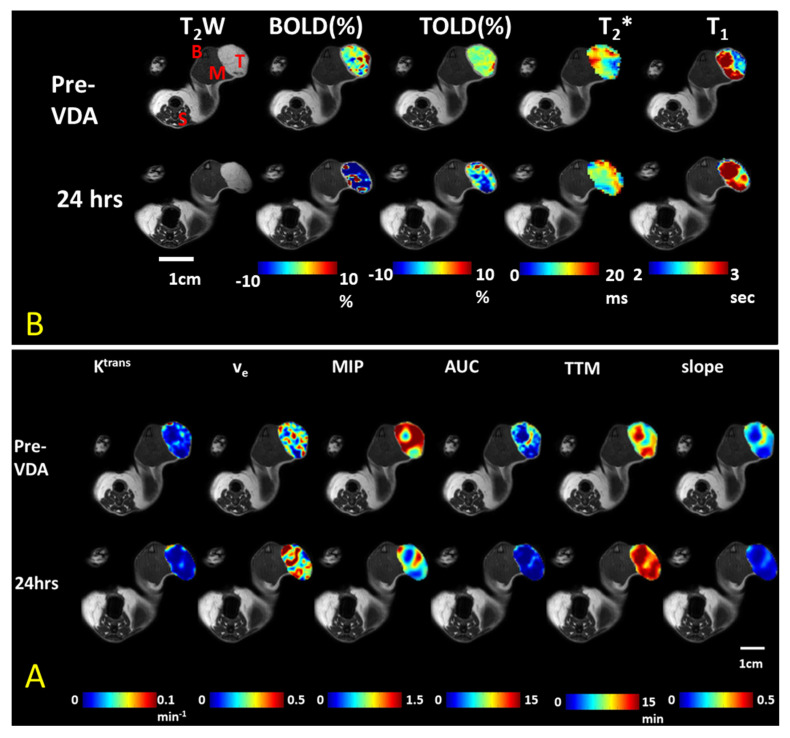
Multiparametric MRI assessment of ischemia and hypoxia in A549 human lung cancer in a rat model. The combretastatin analog OXi6197 was used as an experimental VDA in an SC model in the leg [159]. Labels shown on T_2_W. T-Tumor; M-Muscle; B-Bone marrow; S-Spine; T_2_* is a physical variation on T_2_. (**A**) DCE confirms decreased perfusion based on area under the curve (AUC), time to maximum (TTM) and slope of DCE contrast curves following dose of 15 mg/kg OXi6197; (**B**) Oxygen-sensitive parameters (BOLD and TOLD) show much smaller response in vascular oxygenation to an oxygen gas breathing challenge after VDA. BOLD is sensitive to the amount of deoxyhemoglobin and has considerable similarity to the MSOT measurements (Figure 8, Figure 9 and Figure 10). Data in B were acquired prior to A.

**Figure 8 molecules-26-02551-f008:**
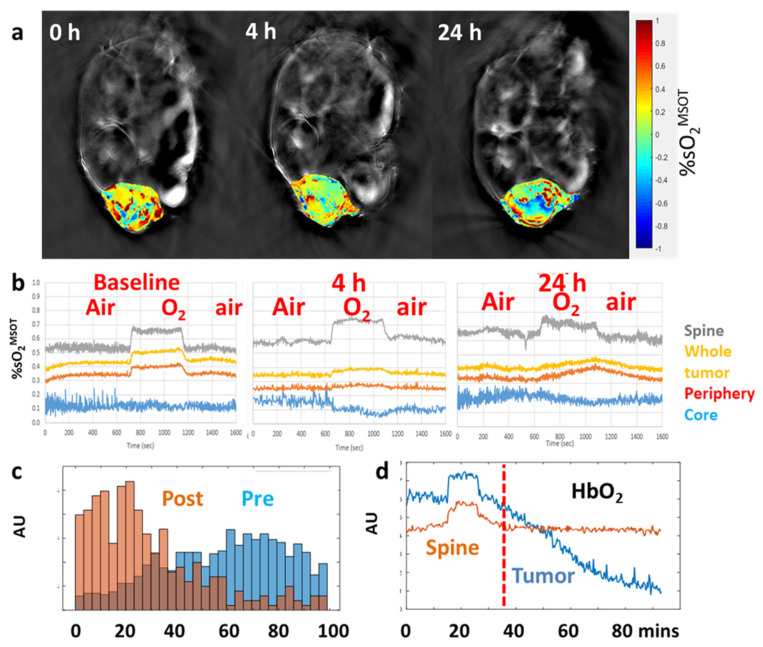
MSOT reveals progression of vascular shutdown. (**a**) Trans-axial MSOT images of nude mice with MDA-MB-231-luc breast tumor. Color oxygenation maps overlaid on single wavelength anatomical grayscale images. Progression of hypoxiation in a mouse treated with CA4P (120 mg/kg), with strong response to treatment (white arrows) resulting in vascular impairment within 4 h. (**b**) Traces show vascular oxygen saturation response (sO_2_^MSOT^) to O_2_-breathing challenge at baseline, 4 and 24 h after CA4P. Yellow (whole tumor), red (tumor periphery), blue (tumor center), grey (spine, which serves as effective control tissue). At 4 h, the response of the tumor periphery was depressed and at 24 h, it showed a very different (sluggish) pattern. (**c**) Histogram of vascular oxygen saturation in tumor periphery before and after CA4P in a second tumor bearing mouse; (**d**) Traces show concentration of HbO_2_ before CA4P and over 1 h following CA4P IP.

**Figure 9 molecules-26-02551-f009:**
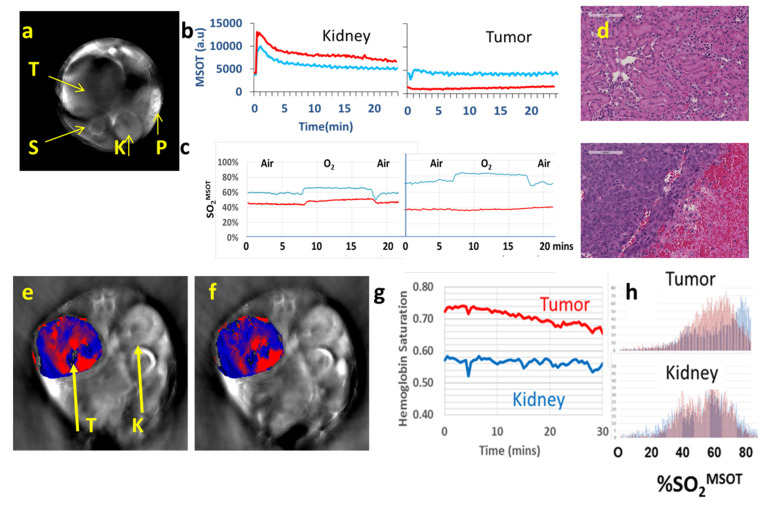
Photoacoustic assessment of vascular disruption in kidney tumors. (**a**) MSOT of orthotopic RENCA tumor in BALB/C mouse. Transaxial 800 nm MSOT image showing tumor (T), spine (S), contra lateral kidney (K) and spleen (P). (**b**) Dynamic contrast enhanced MSOT accompanying IV infusion of the blood pool agent Genhance. Similar curves were observed at baseline (blue) and 24 h (red) in kidney. In tumor, inflow of Genhance was observed at baseline, but much less at 24 h commensurate with vascular shutdown matching. Blue baseline and red 24 h after OXi8007 (350 mg/kg). (**c**) Traces showing hemoglobin oxygen saturation derived from dynamic MSOT images in spine and tumor accompanying oxygen-breathing challenge. Traces for the tumor showing baseline response, but no activity after 24 h indicating vascular shutdown. Area under the curve indicates about 95% less signal. (**d**) H&E stained section of RENCA tumor (lower panel) and contralateral kidney (upper panel) from resected tissue obtained 72 h after OXi8007. Extensive hemorrhage is seen in the tumor. (**e**) Acute response of human RCC XP373 to CA4P. Distinct anatomy is apparent in the transaxial slice MSOT image showing tumor (T) and contralateral control kidney (K). Red indicates predominant oxyhemoglobin and blue deoxyhemoglobin in tumor at baseline. (**f**) Regional hypoxiation is seen in the tumor vasculature following CA4P. (**g**) Dynamic changes were observed in tumor over 30 min following CA4P (120 mg/kg, IP), while contralateral normal kidney showed no change as confirmed in (**h**) histograms, verifying selective activity against the tumor.

**Figure 10 molecules-26-02551-f010:**
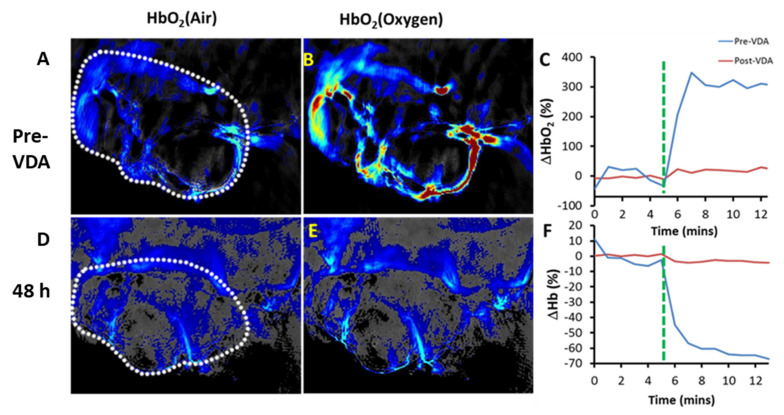
Photoacoustic imaging showed increase of oxyhemoglobin from air (**A**) to oxygen breathing (**B**) at baseline (tumor outlined in **A** and **D**). (**D**,**E**) Two days after the administration of VDA OXi6197 only minimal response to oxygen was observed. (**A**, **D**: air breathing; **B**, **E**: oxygen breathing). Comparison of normalized (**C**) oxy-(ΔHbO_2_) and (**F**) deoxyhemoglobin (ΔHb) at both time points. These data closely match the MRI obtained as part of the same study (Figure 7). Part of this Figure adapted from [175].

**Figure 11 molecules-26-02551-f011:**
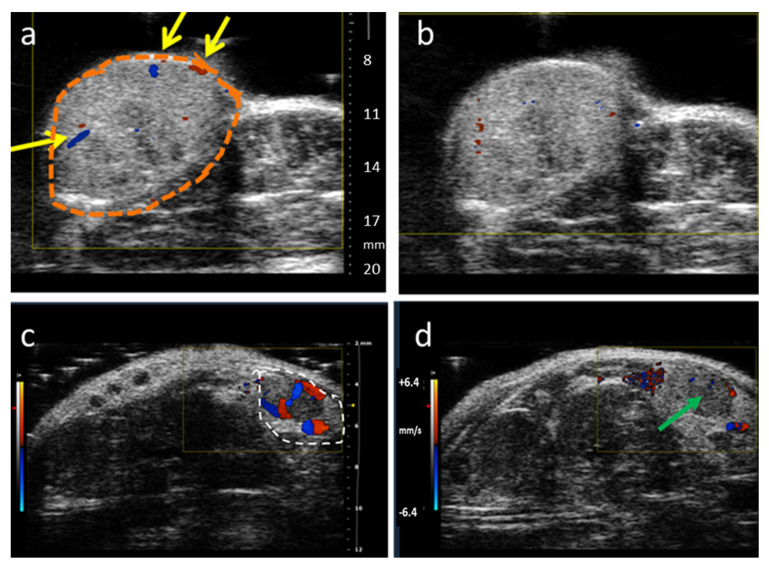
Color-Doppler ultrasound of tumor vascular oxygenation. A549 human lung tumor xenograft growing in hind leg of rat was imaged (**a**) before and (**b**) 90 min after administration of OXi6197 IP (15 mg/kg). A catheter with solution of OXi6197 was secured IP to avoid moving the animal during the experiment. Tumor outlined in orange, yellow arrows indicate regions of substantial flow, which was initially very sparse and diminished following administration of VDA. Transmit Frequency: 16 MHz Similar study of subcutaneous human prostate tumor PC-3 xenograft in a SCID mouse observed (Transmit Frequency: 24 MHz) (**c**) immediately after and (**d**) about 80 min after injection of OXi8007 (350 mg/kg IP). This tumor was initially far better vascularized, but tumor perfusion was essentially halted by 80 min (green arrow), as published for other time points previously [74]. Heat scale bar representing flow in the range ±64.2 mm/s. Both investigations were performed using a Vevo 2100.

**Table 1 molecules-26-02551-t001:** Vascular disrupting agents. Status and use of imaging during recent clinical trials [19,25].

Agent	Imaging Modality	Tumor Type	Trial	References
CA4P (fosbretabulin; Zybrestat)	DCE-MRI or -CT ^a^ DWI-MRI ^b 15^O-PET	Lung cancer, ovarian, renal, breast	Phase 1	[26,27,28,29] ^a^ [30] ^b^ [31]
CA1P (OXi4503)	DCE-MRI ^15^O-PET	Various	Phase 1	[32]
BNC105P	DCE MRI	Various	Phase 1 21 patients; now in Phase 2	[33]
CYT997	DCE-MRI	Various	31 patients	[34]
AVE8062 (Ombrabulin)	DCE-US	Various, mostly ovarian	25 patients	[35]
NPI2358 (Plinabulin)	DCE-MRI	Various	38 patients	[36]
ZD6126	DCE-MRI	Colon		[37]
EPC2407 (Crolibulin)	DCE- & DWI-MRI	Various	11 subjects, Phase 1	[38]
MN-029 (Denibulin)	DCE-MRI	Various	34 subjects, Phase 1	[39]
DMXAA (ASA404; 5,6-dimethylxanthenone-4-acetic acid; vadimezan)	DCE-MRI	Various	Phase 1	[40]

Abbreviations. DCE: dynamic contrast enhanced; DWI: diffusion weighted imaging; MRI: magnetic resonance imaging, CT: computed tomography; US: ultrasound; PET: positron emission tomography.

**Table 4 molecules-26-02551-t004:** Comparison of imaging modalities.

Modality	Cost	Throughput	Spatial Resolution	Temporal Resolution	Need for Contrast Agent	Ease of Use
BLI	$	High 5–10 mice	Surface planar	1 s	Yes	Easy
MRI	$$$$$	Usually single subject	200 µm in plane × 2 mm	s-mins	Typically, yes	Sophisticated methods available
MSOT	$$$	Single subject	120 µm in plane by 200 µm	s	No	Image quality very sensitive to meticulous setup
US	$$	Single subject	100 to 1000 µm	Sub second	Often	Fairly easy
PET/CT	$$$$$	Typically, 1–4 mice or single larger subject	3 mm isotropic	mins	Yes	Issues of radioactivity: expense/safety

## Data Availability

The data presented in this study are available on request from the corresponding author.

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
