# Peer review of "Non-Invasive Evaluation of Acute Effects of Tubulin Binding Agents: A Review of Imaging Vascular Disruption in Tumorsâ€"

_molecules, 2021, doi:10.3390/molecules26092551_

Round 1
Reviewer 1 Report
In this review the authors first summarize the nature and mechanisms of action of several vascular disrupting agents (VDAs), many of which affect tubulin physiology, and the pass to describe methods to analyze VDAs effects on cancers invite in animal models. Although in the opinion of this reviewer the methods described do not clearly distinguish the effects of drugs on tumor cells rather than on tumor vascularization, the review is clearly written, leaves space to potential different interpretation of the imaging results and therefore useful to readers.
Author Response
Thank you for the positive assessment of the Review; it appears no specific changes are required. We will thoroughly spell check again.
Reviewer 2 Report
This is a very interesting and comprehensive review on the use of imaging biomarkers to assess the effect of vascular disrupting agents. The paper is very well written, structured and illustrated by figures showing the potential of different imaging strategies.
I have not found any error or missing technical information. The paper is also interesting because it will be a source of inspiration for researchers working in other studies and/or pathologies where haemodynamics is crucial.
My sole suggestion would be to add a short perspective on the use of imaging to guide combination of VDAs with other drugs. It seems that imaging may also play an important role in this matter to predict a potential benefit (or not) of this type of combination . Suggestion of references:
- Combretastatin A4 Nanoparticles Combined with Hypoxia-Sensitive Imiquimod: A New Paradigm for the Modulation of Host Immunological Responses during Cancer Treatment. Shen N, et al. Nano Lett. 2019 Nov 13;19(11):8021-8031
- The Blood Flow Shutdown Induced by Combretastatin A4 Impairs Gemcitabine Delivery in a Mouse Hepatocarcinoma. Fruytier AC, et al. Front Pharmacol. 2016 Dec 23;7:506.
- A cooperative polymeric platform for tumor-targeted drug delivery. Song W, et al. Chem Sci. 2016 Jan 1;7(1):728-736.
- Vascular priming enhances chemotherapeutic efficacy against head and neck cancer. Folaron M, et Oral Oncol. 2013 Sep;49(9):893-902.
- Vascular-disrupting agents in oncology. Mita MM, et al. Expert Opin Investig Drugs. 2013 Mar;22(3):317-28
Author Response
Thank you for the suggested additional references. We have now added a subsection 2.5 as follows and shown on the tracked version of manuscript.
2.5. Optimizing combination therapy
As mentioned in the Introduction, effective treatment with VDAs will likely require combination therapy, specifically to overcome the peripheral surviving tumor tissue ring observed in most studies [11, 41, 56, 251]. Imaging should be particularly effective in facilitating optimal combination based on timing and extent of acute vascular changes. Notably, enhanced permeability may promote delivery of additional drugs, though ischemia may limit access. Consequent hypoxia is expected to reduce the efficacy of radiation therapy, though could promote hypoxia activated pro-drugs [252].
Building on earlier DCE MRI evaluations of tumor perfusion following VDA treatment [106], Fruytier et al. established that delivery of gemcitabine was diminished in TLT hepatocarcinomas growing intramuscularly in mice 2 hrs after CA4P. They elegantly showed changes in vascular perfusion and permeability based on DCE-MRI, as well as assessing uptake and metabolic conversion of gemcitabine using 19F NMR spectroscopy [107]. Such an approach could allow effective determination of optimal timing of combined therapy by establishing both the pharmacokinetics and uptake of a particular therapeutic agent, as well as the pharmacodynamic vascular perturbation caused by the VDA.
Folaron et al. examined “vascular priming” to enhance the efficay of several common chemotherapeutic drugs through combination with DMXAA. They specifically applied dynamic BLI to examine tumor growth and DCE-MRI to evaluate changes in tumor perfusion and permeability in relation to the efficy of irinotecan, docetaxel, and doxorubicin [134]. Intriguingly, they found increased BLI signal 1 hr after DMXAA suggesting enhanced delivery of luciferin substrate, but significantly diminished signal at 24 hrs. consistent with vascular collapse, as confirmed by DCE-MRI.
Delivery of chemotherapeutic agents can be enhanced through active targeting and encapsulation in nanoparticles to improve local retention. Sun et al. designed a “cooperative polymeric platform” for tumor-targeted drug delivery. Recognizing that the peptide GNQEQVSPLTLLKXC (A15) is a substrate of activated blood coagulation factor XIII (a transglutaminase), they created A15 peptide-decorated poly(L-glutamic acid)-cisplatin conjugates as coagulation-targeted nanoparticles [140]. They then exploited the VDA DMXAA to induce hemorrhage in tumors yielding “a unique coagulation environment”. Using MSOT they were able to convincingly show elevated uptake of such fluorescently (NIR830)-labeled NPs after administration of DMXAA and ultimately demonstrate enhanced tumor growth delay in mice.
Zhao et al. showed distinct hypoxiation of 13762NF rat breast tumors within 30 mins of 30 mg/kg CA4P using near infrared spectroscopy and additionally found that the tumor became essentially unresponsive to an oxygen gas breathing challenge at 2 hrs, although some response was restored after 24 hrs [253]. This coincided with greatest tumor growth delay being achieved when tumors were irradiated (5 Gy), while rats breathed oxygen 24 hrs after CA4P, whereas other sequences of treatment were less effective. Diepart et al. used 19F MRI to determine that tumor pO2 increased transiently for about 2 hrs after administration of ATO (5 mg/kg) to mice [80]. This suggested tumor irradiation at 90 mins following ATO would be particularly effective, as indeed observed.
VDA induced hypoxia has also been explored to promote activation of bioreductively-activated prodrugs [148, 254]. Notably, Shen et al. developed a paradigm for modulation of host immunological responses during cancer treatment by exploiting CA4 and the immune modulator Imiquimod (IMQ) [254]. Noting that TIE2+ tumor-associated macrophages (MΦ) and endothelial progenitor cells have been reported to infiltrate tumors after treatment with CA4P, thereby promoting tumor angiogenesis, [255], it was reasoned that the immune modulator IMQ could potentially convert immature plasmacytoid dendritic cells (pDCs) into their active form, leading to the robust infiltration and priming of natural killer cells and cytotoxic T-lymphocytes in treated tumors. To seek tumor specificity a bioreducible prodrug hs-IMQ was prepared and co-administered with CA4 in poly(l-glutamic acid)-graft-methoxy poly(ethylene glycol nanoparticles (NPs). The NP depots ensured longer-term delivery/release of CA4, which induced additional tumor hypoxia promoting nitroreductase activity and IMQ release. Therapeutic efficacy in 4T1 breast tumors in mice was enhanced. Pimonidazole was also administered to reveal increased hypoxia in tissue slices commensurate with release of IMQ. In vivo imaging was not applied, but this study represents an ideal opportunity for applying 18F-miso PET, oxygen-sensitive MRI or MSOT to examine the dynamic evolution of hypoxia non-invasively.